

# On population structure and breeding biology of burrowing crab *Dotilla blanfordi* Alcock, 1900

Krupal Patel[1,*], Heris Patel[1,*], Daoud Ali[2], Swapnil Gosavi[3], Nisha Choudhary[1], Virendra Kumar Yadav[1], Kauresh Vachhrajani[3], Ashish Patel[1], Dipak Kumar Sahoo[4] and Jigneshkumar Trivedi[1]

[1] Department of Life Sciences, Hemchandracharya North Gujarat University, Patan, Gujarat, India
[2] Department of Zoology, College of Science, King Saud University, Riyadh, Saudi Arabia
[3] Department of Zoology, The Maharaja Sayajirao University of Baroda, Vadodara, Gujarat, India
[4] Department of Veterinary Clinical Sciences, College of Veterinary Medicine, Iowa State University, Ames, IA, United States of America
[*] These authors contributed equally to this work.

## ABSTRACT

**Background**. The present study investigated the population structure and breeding biology of the burrowing brachyuran crab species *Dotilla blanfordi* Alcock, 1900, which is commonly found on the sandy beach of Bhavnagar, located on the Gulf of Kachchh, Gujarat coast, India.

**Methods**. Monthly sampling was conducted from February 2021 to January 2022 at the time of low tide using three line transects perpendicular to the water line, intercepted by a quadrate ($0.25 \text{ m}^2$) each at three different levels of the middle intertidal region: 20 m, 70 m, and 120 m. The quadrate area was excavated up to 30 cm and sieved for specimen collection. The collected specimens were categorised into different sexes viz., male, non-ovigerous female, or ovigerous female. For the fecundity study of *D. blanfordi*, the carapace width (mm) as a measure of size as well as their wet weight (g), size, number, and mass of their eggs were also recorded.

**Results**. The study revealed sexual dimorphism among the population, with females having significantly smaller sizes as compared to males. The overall population was skewed towards females, with a bimodal distribution of males and females. The occurrence of ovigerous females throughout the year suggests that the population breeds incessantly throughout the year, with the highest occurrence in August and September. A positive correlation was observed between the morphology of crabs (carapace width and wet body weight) and the size, number, and mass of eggs.

# INTRODUCTION

Studies conducted on population structure and breeding biology primarily try to understand the way individuals are distributed within a population with respect to several aspects of its survival and growth (*Saher & Qureshi, 2010*; *Saher & Qureshi, 2011*; *Hu et al.,*

Corresponding authors
Dipak Kumar Sahoo, dsahoo@iastate.edu
Jigneshkumar Trivedi, jntrivedi26@yahoo.co.in

*2015*; *Manzoor et al., 2016*). In this way, researches have demonstrated that most tropical and subtropical crab species breed continuously (*Litulo, 2004*; *Bezerra & Matthews-Cascon, 2007*; *Manzoor et al., 2016*), while species occurring in temperate regions show seasonality in breeding patterns (*Carmichael, Rutecki & Valiela, 2003*; *Bas, Luppi & Spivak, 2005*; *Cartwright-Taylor, Lee & Hsu, 2009*). At the same time, intraspecific differences in peak reproductive activity among the populations are common, being this a response to the variation in biotic and abiotic factors of the habitat they occupy (*Negreiros-Fransozo, Costa & Colpo, 2003*; *Litulo, 2004*). For this reason, estimating population fecundity and sexual maturity is crucial in predicting the turnover capacity of natural populations and understanding the long-term impact of the environment (*Mantelatto & Fransozo, 1997*; *Pinheiro, Freire & Lins-Oliveira, 2003*).

The crabs of the family Dotillidae are one of the most familiar amphibious and terrestrial crabs found on sandy-muddy shores. Among these crabs, *Dotilla* crabs, commonly known as soldier or bubbler crabs, are one of the important ecological agents of the intertidal zone of tropical and sub-tropical regions (*Manzoor et al., 2016*). These crabs are well known for making burrows in the sandy or muddy intertidal region and making feeding pellets (*Desai et al., 2022*; *Joshi, Patel & Trivedi, 2022*; *Upadhyay et al., 2022*). On the taxonomical aspect, numerous studies are available on the species of infraorder Brachyura and Anomura (*Trivedi & Vachhrajani, 2015*; *Trivedi & Vachhrajani, 2018*; *Trivedi, Gosavi & Vachhrajani, 2020a*; *Trivedi et al., 2020b*; *Trivedi et al., 2021*; *Gosavi et al., 2021*; *Patel, Patel & Trivedi, 2020*; *Patel, Patel & Trivedi, 2021*; *Bhat & Trivedi, 2021*; *Padate et al., 2022*). However, only a few studies are available on the ecological aspects of brachyuran and anomuran crabs occurring along the Gujarat coast (*Trivedi & Vachhrajani, 2016*; *Trivedi & Vachhrajani, 2017*; *Patel, Vachhrajani & Trivedi, 2022*; *Patel et al., 2024*).

Out of various species of brachyuran crabs reported from the mudflat habitat of Gujarat State (*Gosavi et al., 2021*), *Dotilla blanfordi Alcock, 1900* is the most common burrowing crab species. In India, the species is found throughout most of the coastal states, including Gujarat, Maharashtra, Andhra Pradesh, Odisha, and West Bengal (*Trivedi et al., 2018*). *Dotilla blanfordi* is well known to construct burrows in soft sediments and feed on organic matter deposited on the sediment surface by the formation of pseudofecal pellets. As a result of its unique feeding behaviour, the species act as ecosystem engineer which helps in the bioturbation of the uppermost layer of the sediment up to a few millimetres (*Litulo, Mahanjane & Mantelatto, 2005*). The species have dense populations and have a significant impact on the ecological aspects of mudflat habitats as a result of their burrowing and feeding habits (*Allen, 2010*). A total of three species of *Dotilla* are reported from Gujarat, including *D. malabarica Nobili, 1903*; *D. myctiroides Milne Edwards, 1851*; and *D. blanfordi*. Among these, *D. blanfordi* is abundantly found on the mudflats of the Gulf of Khambhat. The species is distributed from Jafarabad to Umargam on the Gulf of Khambhat region (*Desai et al., 2022*; *Joshi, Patel & Trivedi, 2022*; *Upadhyay et al., 2022*), with a higher abundance at Kuda Beach in Bhavnagar district, Gujarat state, India.

Understanding the population structure and breeding biology of a species can provide valuable insights about the age structure of the population, which impacts its stability and growth rate (*Manzoor et al., 2016*); the spatial arrangement of individuals, which affects
resource distribution (*Sant'Anna et al., 2006*); and the ability to reproduce successfully, which is crucial for survival and persistence (*Bertini, Fransozo & Braga, 2004*). Additionally, the reproductive output of a population is a critical factor that determines its growth and dynamics, making it possible to predict changes in size and structure over time (*Childress, 1972*). Fecundity as well as the pace of generation of broods are unique characteristics that may differ in different populations of a species (*Cody, 1966*). Among these characters, fecundity can be the best indicator of the reproductive fitness of an individual (*Sastry, Vernberg & Vernberg, 1983*), which can help in understanding how reproduction strategies are linked with environmental factors (*Cody, 1966*; *Alkhafaji et al., 2017*). Similar studies elucidating population trends with respect to environmental factors on dotillid crabs occurring on the Gujarat coast have not been carried out till now. The present research was carried out under the hypothesis that the population structure of *D. blanfordi* has an ideal sex ratio (1:1) with no disparities between the sexes and no effect of temperature on the fecundity of the species. The aim of testing the hypothesis was (1) to understand the population structure and (2) to understand the reproductive biology of *D. blanfordi*. The study would yield baseline data essential in comprehending the impacts of shifting environments, conditions, or anthropogenic activities. The current investigation will also contribute in clarifying the research area's coastal health.

## MATERIAL AND METHODS

### Study area

The present study was conducted on the muddy/sandy beach of Kuda Beach (21° 37′31.1″N, 72° 18′19.1″E), Bhavnagar City, Gulf of Khambhat Coast, Gujarat (Fig. 1). The study site falls in a region with a hot semi-arid climate that experiences three seasons in a year: winter (November–December), summer (March–June) and monsoon (July–October) (*Rao & Rama Sarma, 1990*). The study site is located in the Gulf of Khambhat (GoKh) region of the Gujarat coast, which is a funnel-shaped indentation (Fig. 1). With an average tidal range of 5.82 m and a maximum range of 11.66 m, it is the highest tidal range in the Arabian Sea and the second largest measured anywhere in the world (*Mitra, Kumar & Jena, 2020*). The stretch of the beach is around 4 km long, with the upper region having coarse sandy sediment and the lower intertidal region having muddy substratum. The anthropogenic activity is very minimal on the study site which provides a perfect study place for the population structure and breeding biology of *D. blanfordi*.

### Field methods

Regular month-wise sampling was conducted for 12 consecutive months (February 2021 to January 2022). During low tide, the region dominated by *D. blanfordi* was first identified for sampling since the species is not distributed throughout the intertidal region. The burrows constructed by *D. blanfordi* can be easily recognised by their specific mud ball pattern present around the burrow opening and/or the presence of a chimney (Fig. 2).

During the present study, *D. blanfordi* population was found to be distributed in the middle intertidal region (20–120 m) of the study area, where the transects were laid

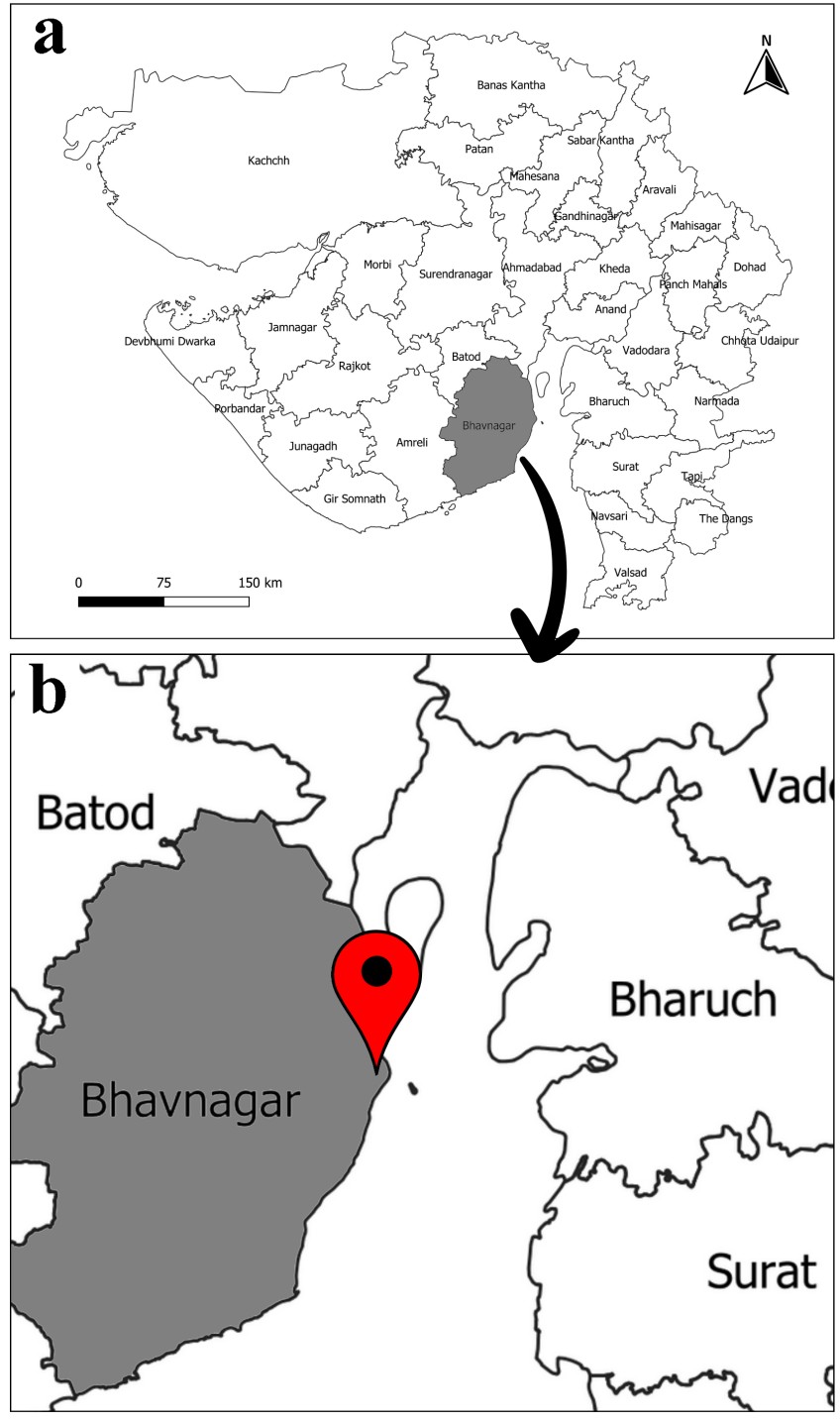

**Figure 1  Map of study area.** (A) Gujarat state, India, (B) Kuda, Bhavnagar, Gulf of Khambhat, Gujarat. (Map constructed using QGIS version 3.14).

perpendicular to the shoreline. A total of three transects were laid, which were 100 m apart from each other. On each transect, three quadrates (0.25 m$^2$) were laid at three different

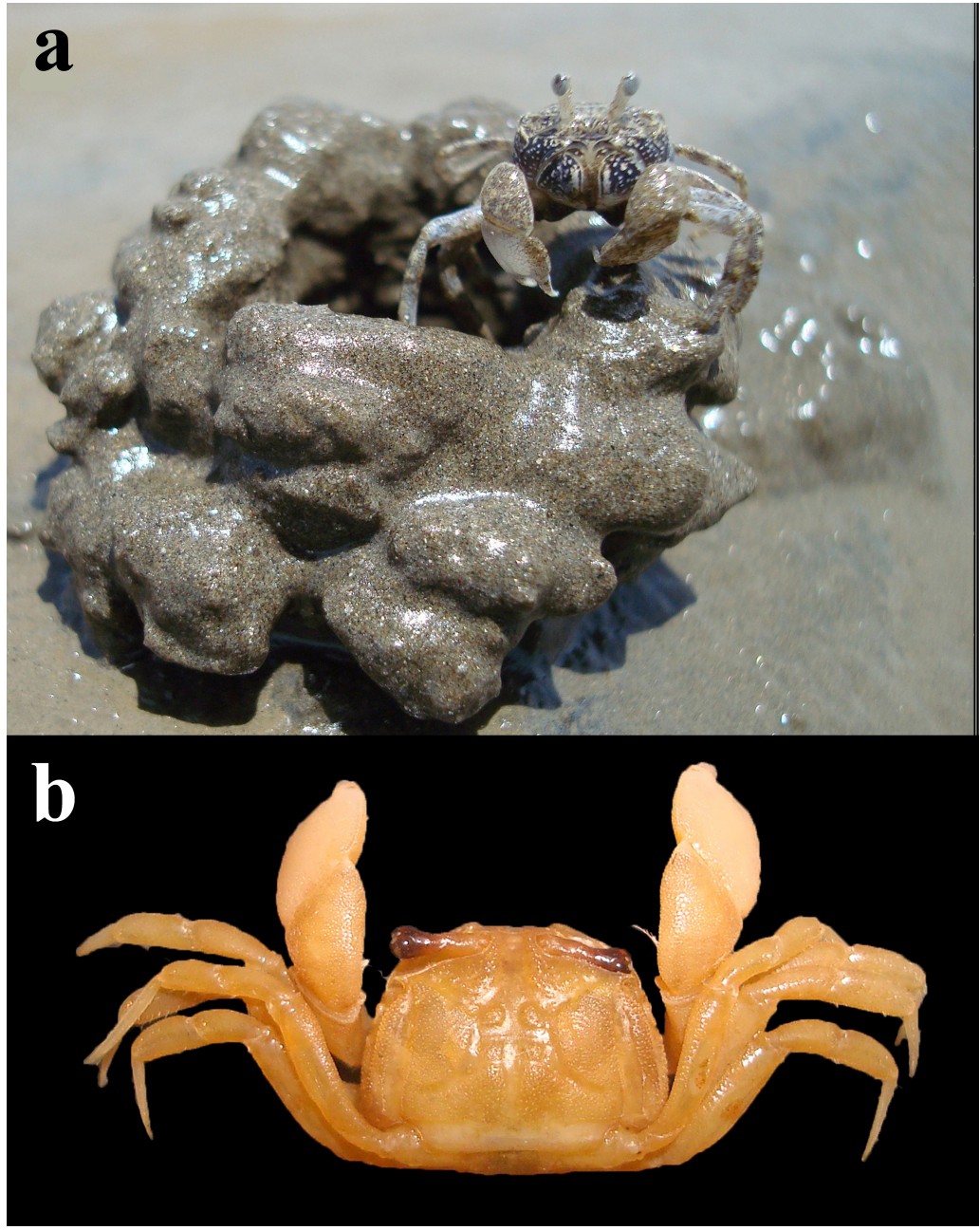

**Figure 2** *Dotilla blanfordi* Alcock, 1900 (CW: 3.54 mm) (A) male specimen in habitat; (B) dorsal view.

levels of the middle intertidal region: 20 m, 70 m, and 120 m (Fig. 3). The quadrate area was excavated up to 30 cm since the species does not burrow beyond that depth, and the soil was sieved (1 mm mesh) (adopted from *Litulo, Mahanjane & Mantelatto (2005)*). The collected samples were kept in pre labelled zip-lock bags, placed in an icebox, and carried to the laboratory. Moreover, the ambient (air) temperature was recorded each month during sampling time. The ambient temperature was recorded using a digital thermometer.

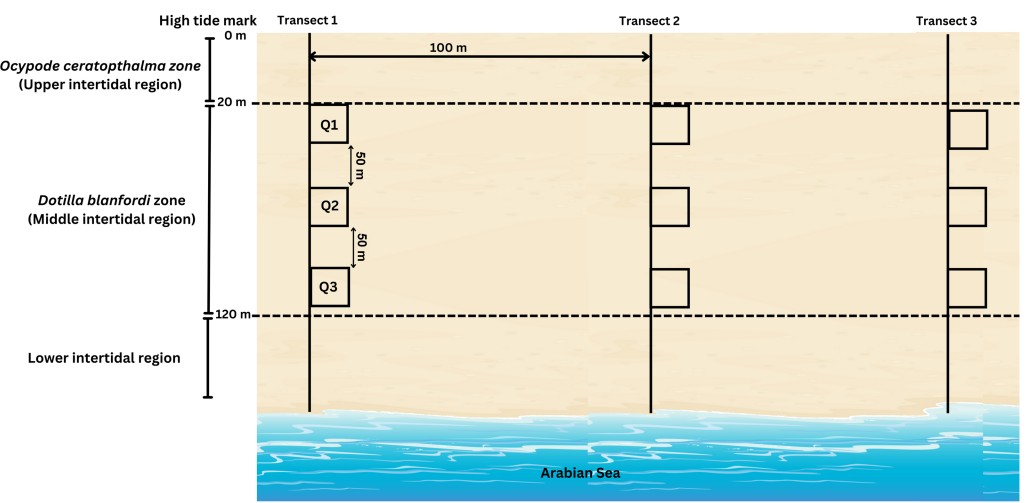

**Figure 3** Transect intercepted by quadrates method carried out for sample collection of *D. blanfordi* from the study site.

## Laboratory analyses

The specimens were preserved in 10% formalin for further analysis in the laboratory. Identification of crabs was first confirmed using standard keys (*Allen, 2010*), and the individuals were categorised as male, non-ovigerous female, or ovigerous female (egg carrying female). For morphological characteristics, the carapace width (CW) was measured using vernier callipers (with ±0.01 mm accuracy), and the wet body weight (BW) (soon after specimen collection) of every individual was measured with a weighing scale (0.001 g accuracy). Individuals that were smaller than the smallest ovigerous female from the collected samples were considered juvenile (<3.28 mm CW) (*Mantelatto & Garcia, 2001*; *Baeza et al., 2013*).

For fecundity studies, a total of 40 ovigerous females were selected randomly from the collected samples ($n = 40$). Fecundity was assessed in terms of total number of eggs, egg mass weight, and size of eggs (diameter) in ovigerous females. The ovigerous females were weighed first, along with the egg mass. Later, the egg mass was carefully removed and weighed again, and the difference was considered as the egg mass weight. For the total number of eggs, egg mass was carefully removed prior to fixation with the help of a brush and forceps from the pleopods of ovigerous females, transferred to 20 ml of sea water, and mixed gradually so that the eggs did not break and were evenly distributed. To know the total number of eggs in the above solution, three samples of 2 ml each were taken in a petri dish and examined under a stereomicroscope. The average of the total number of eggs counted from the three samples was multiplied by the dilution factor (DL = 10) (*Litulo, 2004*) for an estimation of the total number of eggs. The egg diameter ($n = 10$) of each ovigerous female was measured using an ocular micrometre attached to a stereo microscope (*Saher & Qureshi, 2010*).

## Data analyses

The analysis of the population size structure was done in relation to the individual size frequency distribution. The sampled specimens ranged from 1 to 8 mm CW in size; hence, the individuals were grouped in a 1 mm size class interval. The Shapiro–Wilk test was performed to determine the normality of the obtained data, and the results indicated that the data was not normally distributed ($p < 0.001$), hence non-parametric techniques were employed. The Kruskal-Wallis (KW) test was used to examine the variation in the mean values of CW between the sexes of *D. blanfordi* (males, non-ovigerous females, and ovigerous females) in order to determine the degree of sexual dimorphism. When the KW test revealed a significant difference ($p < 0.005$), for multiple comparison testing, Dunn's *post hoc* test was carried out.

The monthly plot of sex composition and size variations was prepared using the monthly data of crab size (CW) and sex (modal distribution). The peak of the breeding season is indicated by more ovigerous females in the monthly samples. The chi-square test ($\chi 2$) was calculated to evaluate the sex ratio between male and female (non-ovigerous females and ovigerous females) individuals. In order to determine the size of females at first maturity, the proportion of ovigerous females in each size class was calculated for the entire year, spanning from March 2021 to February 2022. To obtain a better understanding of how temperature affects *D. blanfordi* mating and juvenile settlement, monthly data on the relative frequency of juvenile and ovigerous females was plotted against ambient temperature. Furthermore, Pearson's correlation analysis was used to understand the relationship between the average ambient temperature and the relative abundance of juveniles. The relationship between the morphological characteristics of eggs and the morphology of *D. blanfordi* crabs (CW and BW) was estimated using a regression analysis. If $p < 0.05$, statistical significance was recognised. Microsoft Excel and PAST software, version 4.03, were used to carry out all of the statistical analyses.

## RESULTS

*Dotilla blanfordi* (Fig. 2) showed a patchy distribution and was mostly found in the middle intertidal region (20 m to 120 m vertical length of the beach), while the upper intertidal region was dominated by the species *Ocypode rotundata*. The average abundance of *D. blanfordi* in the middle intertidal region was recorded to be $20.44 \pm 42.24$ individuals /m$^2$. A total of 1,696 *D. blanfordi* individuals were collected, including 904 (53.30%) males, 571 (33.67%) non-ovigerous females, and 221 (13.03%) ovigerous females (Table 1). *Dotilla blanfordi* CW ranged between 1.07 mm and 7.34 mm, with the average size of males, non-ovigerous females and ovigerous females being $4.11 \pm 1.26$ mm, $3.70 \pm 0.95$ mm, and $4.12 \pm 0.49$ mm, respectively. We observed prominent sexual dimorphism with male individuals ($4.11 \pm 1.26$ mm) being significantly larger (Kruskal-Wallis, $H = 16.69$, $p < 0.001$) as compared to female individuals ($3.81 \pm 0.86$ mm). Dunn's *post hoc* tests revealed that, among female individuals, ovigerous females were significantly larger than non-ovigerous females (Dunn's test, Bonferroni corrected, $p < 0.001$), while males were significantly larger than females (Table 1).

**Table 1  Carapace width of different sexes of *Dotilla blanfordi* individuals.**

| Sex | n | Min CW(mm) | Max CW(mm) | Mean ± SD |
|---|---|---|---|---|
| Male | 904 | 1.07 | 7.34 | 4.11 ± 1.26*** |
| Non-ovigerous female | 571 | 1.14 | 6.66 | 3.70 ± 0.95*** |
| Ovigerous female | 221 | 3.27 | 5.36 | 4.08 ± 0.46*** |

Notes.

Kruskal–Wallis; *n*, total individuals; CW, carapace width.

***$p = < 0.001$.

**Table 2  Total number of *Dotilla blanfordi* specimens collected from Kuda Beach, Bhavnagar.**

| Month | M | % | NOF | % | OF | % | NOF+OF | % | Male:Female |
|---|---|---|---|---|---|---|---|---|---|
| January | 85 | 55.19 | 53 | 34.42 | 16 | 10.39 | 69 | 44.81 | 1:0.8 |
| February | 66 | 62.26 | 31 | 29.25 | 9 | 8.49 | 40 | 37.74 | 1:0.6 |
| March | 66 | 48.53 | 60 | 44.12 | 10 | 7.35 | 70 | 51.47 | 1:1.1 |
| April | 96 | 55.49 | 69 | 39.88 | 8 | 4.62 | 77 | 44.51 | 1:0.8 |
| May | 64 | 49.23 | 56 | 43.08 | 10 | 7.69 | 66 | 50.77 | 1:1.03 |
| June | 59 | 56.19 | 40 | 38.10 | 6 | 5.71 | 46 | 43.81 | 1:0.7 |
| July | 24 | 27.27 | 61 | 69.32 | 3 | 3.41 | 64 | 72.73 | 1:2.7 |
| August | 72 | 44.72 | 39 | 24.22 | 50 | 31.06 | 89 | 55.28 | 1:1.2 |
| September | 111 | 59.04 | 20 | 10.64 | 57 | 30.32 | 77 | 40.96 | 1:0.7 |
| October | 95 | 57.93 | 45 | 27.44 | 24 | 14.63 | 69 | 42.07 | 1:0.7 |
| November | 90 | 62.94 | 40 | 27.97 | 13 | 9.09 | 53 | 37.06 | 1:0.6 |
| December | 76 | 51.35 | 57 | 38.51 | 15 | 10.14 | 72 | 48.65 | 1:0.9 |
| Total | 904 | 53.30 | 571 | 33.67 | 221 | 13.03 | 792 | 46.70 | 1:0.88 |

Notes.

M, male; NOF, non-ovigerous female; OF, ovigerous female.

The overall sex ratio of *D. blanfordi* considerably deviated from the predicted 1:1 proportion ($\chi 2 = 3.70$, $P < 0.05$) and was male-biased (1:0.88). A male-biased sex ratio was recorded during the majority of the months, with the exception of March, May, July, and August (Table 2). The occurrence of ovigerous females all year round demonstrates that the species is reproducing continuously, with August and September exhibiting the highest percentage of occurrence (Table 2).

The *D. blanfordi* individuals were collected in several size classes, ranging from 1 mm to 8 mm CW. The size frequency distribution of male individuals showed a bimodal pattern, with the highest peak falling within the 2–3 mm CW and 7–8 mm CW size classes. In the case of females, a bimodal distribution was also observed, with maximum peak occurring at 3–4 mm CW, and 1–2 mm CW size classes (Fig. 4).

It was observed that males had bimodal distribution during the majority of the month, while in the case of non-ovigerous females, unimodal distribution was observed except in May, June, and July, which showed a bimodal distribution. Ovigerous females showed unimodal distribution throughout the year. The occurrence of juveniles was also recorded in all the months of the year (Fig. 5). Pearson's correlation analysis ($r = 0.07$, $p < 0.001$) revealed a positive correlation between the relative frequency of juveniles and the mean

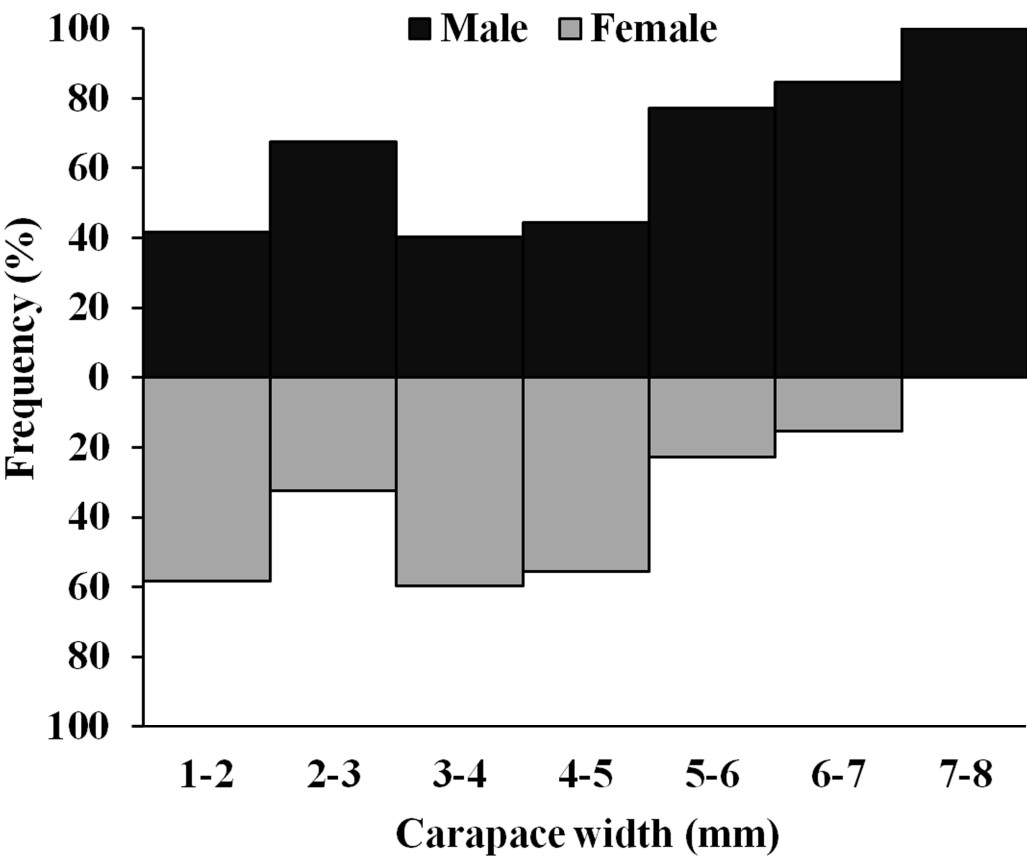

**Figure 4** Overall size frequency distribution of *Dotilla blanfordi* collected from Kuda beach.

ambient temperature. The percentage of ovigerous female and juvenile occurrences (of both sexes) shows no trend with the monthly temperature (ambient and water) (Fig. 6).

The fecundity data revealed the average egg mass weight being $0.02 \pm 0.010$ gm ($n = 40$), the average number of eggs being $715 \pm 309.62$ ($n = 40$), and the average size of eggs being $0.23 \pm 0.042$ mm (Table 3). It was observed that the total egg mass weight, total number of eggs, and size of eggs showed a significant ($p < 0.001$) positive correlation with the CW and BW of the ovigerous females (Fig. 7).

## DISCUSSION

According to the current investigation, the male individuals of *D. blanfordi* were significantly larger as compared to the females. Similar results were obtained for other crabs like *D. fenestrata* (*Litulo, Mahanjane & Mantelatto, 2005*), *D. myctiroides* (*Hails & Yaziz, 1982*), *Matuta planipes* and *Ashtoret lunaris* (*Saher et al., 2017*), and *Scylla olivacea* (*Waiho et al., 2021*). The growth rate of female crabs is generally reduced since they must use their energy for gonadal development, leading to lower somatic growth as compared to males (*Mantelatto et al., 2010*). Moreover, males have greater chances of attracting and
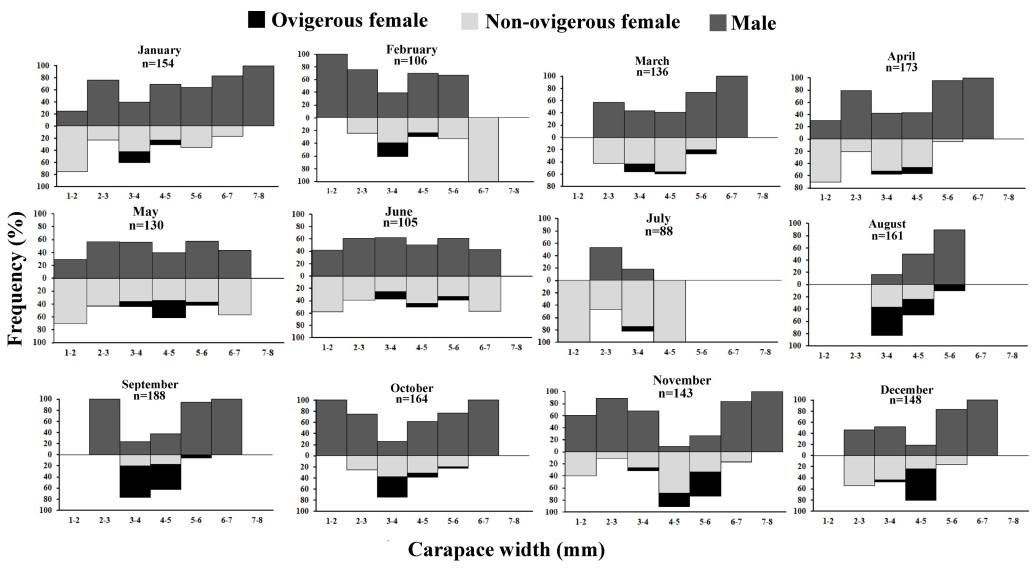

**Figure 5  Monthly size frequency distribution of *Dotilla blanfordi* from January to December.**

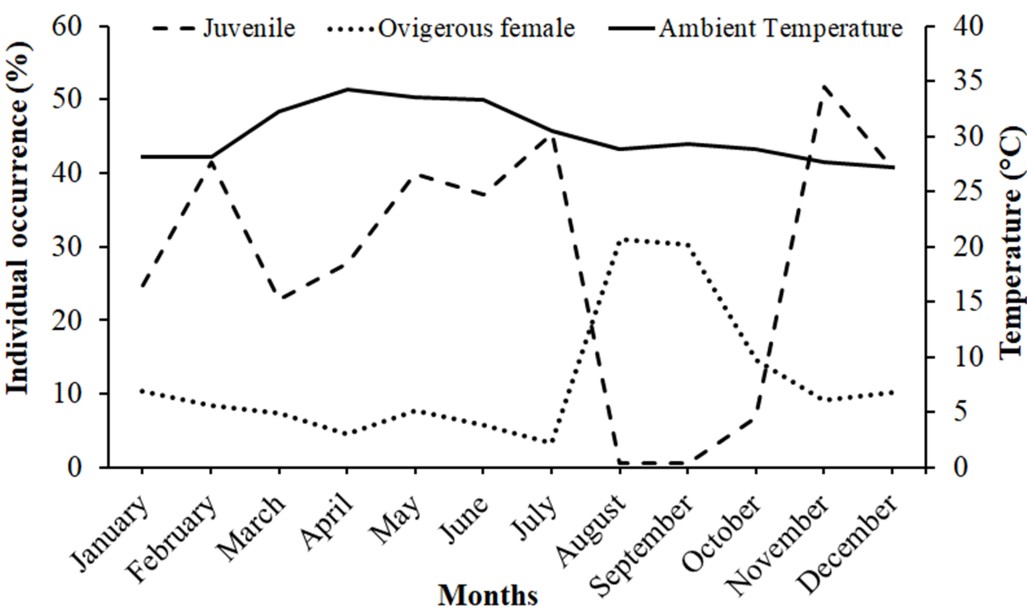

**Figure 6  Association between the juveniles (of both sexes) and ovigerous female occurrence of *Dotilla blanfordi* with monthly ambient temperatures at Kuda Beach.**

mating with females as well as winning intra-specific fights with conspecific males (*Christy & Salmon, 1984*; *Gherardi, Russo & Anyona, 1999*; *Litulo, 2005a*).

During the majority of the months, the monthly sex ratio was skewed towards males. Studies conducted on *D. fenestrata* (*Litulo, Mahanjane & Mantelatto, 2005*), *Scopimera crabicauda* (*Clayton & Al-Kindi, 1998*), *Ocypode rotundata* (*Naderi et al., 2018*), and

**Table 3  Summary of different morphological parameters of *Dotilla blanfordi* ovigerous females and eggs.**

| Variables | n | Mean ± SD | Min. | Max. |
| --- | --- | --- | --- | --- |
| Crab weight (gm) | 40 | 0.03 ± 0.024 | 0.01 | 0.11 |
| Carapace length (mm) | 40 | 3.44 ± 0.54 | 3.09 | 5.6 |
| Carapace width (mm) | 40 | 4.23 ± 0.75 | 2.47 | 4.58 |
| Egg mass weight (gm) | 40 | 0.02 ± 0.010 | 0.004 | 0.05 |
| Egg number | 40 | 715 ± 309.62 | 238 | 1,680 |
| Egg size (mm) | 40 | 0.23 ± 0.042 | 0.13 | 0.35 |

Notes.
n, total individuals.

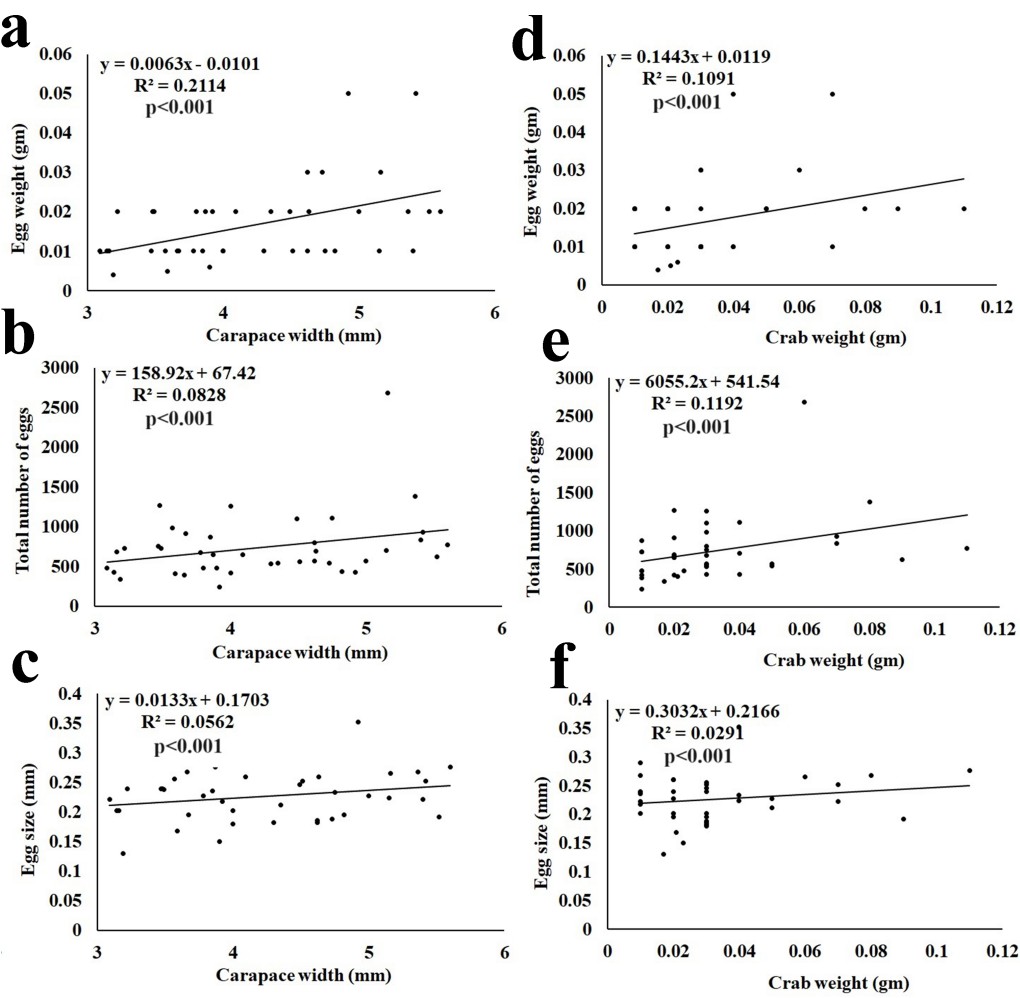

**Figure 7  Relationship of *Dotilla blanfordi* CW with (A) egg mass weight; (B) total number of eggs; and (C) egg size; and BW with (D) egg mass weight; (E) total number of eggs; and (F) egg size.**

*Callinectes sapidus* (*Lycett et al., 2020*) have found similar results. Such a difference in the trends could be the result of seasonal migration patterns, sediment dynamics, and

artefacts during sampling, *e.g.*, depth of sampling (*Litulo, Mahanjane & Mantelatto, 2005*). Moreover, the low ratio of females in the sample sizes could be affected by the behaviour of female individuals of being in their burrows during majority of the time in order to avoid predation, hence making it difficult to capture as compared to male individuals (*Henmi, 2000*).

Moreover, the individuals of smaller size classes (1–3 mm CW) were showing an ideal sex ratio (1:1), while the sex ratio of intermediate individual size classes (3–5 mm CW) was female biased, probably because of the higher mortality rate of male individuals. The larger size classes (5–8 mm CW) were male-biased, as the growth rate of females is slower than that of males and hence could not attain larger size as males (*Mantelatto et al., 2010*). Several other studies obtained similar results where the larger size classes were mostly comprised of males, including *D. blanfordi* (*Manzoor et al., 2016*), *D. fenestrata* (*Litulo, Mahanjane & Mantelatto, 2005*), *D. myctiroides* (*Hails & Yaziz, 1982*), and *Leptuca thayeri* (*Bezerra & Matthews-Cascon, 2007*). *Giesel (1972)* stated that populations that inhabit inconstant environments will have a female-biased sex ratio to maximise the evolutionary potential as a result of unequal selection between male and female individuals. One of the potential causes of the divergence from the optimum sex ratio in various size classes may possibly be sexual dimorphism in size. The greater risk of male death has been cited as the reason why the sex ratio of the intermediate-size class normally tends to be female-biased (*Asakura, 1995*). On the other hand, males attain larger sizes as compared to females in a short period of time, and the sex ratio was mostly male-biased in larger size classes (*Wenner, 1972*). Unbalanced sex ratios in crab populations may also be caused by disparities in sexual mortality and dispersion (*Johnson, 2003*). These biases have been attributed to differences in male and female spatiotemporal distributions, habitat usage patterns, rates of mortality, life spans, feeding preferences, sex longevity, and differential predation on crab sex ratios (*Spivak, Gavio & Navarro, 1991*).

The overall size and frequency distribution of *D. blanfordi* males and females showed a bimodal distribution. The seasonal difference was observed to be significant in terms of the size and frequency distribution of *D. blanfordi*. Bimodal frequency distributions have also been recorded in *D. fenestrata* (*Litulo, Mahanjane & Mantelatto, 2005*), *L. subcylindrica* (*Thurman, 1985*), *O. quadrata* (*Negreiros-Fransozo, Fransozo & Bertini, 2002*), and *D. blanfordi* (*Manzoor et al., 2016*). Variation among the size and frequency distribution of a population can vary throughout due to reproduction and rapid larval recruitment (*Thurman, 1985*). Numerous factors, including differential mortality (*Díaz & Conde, 1989*), differing migratory patterns (*de Arruda Leme & Negreiros-Fransozo, 1998*; *Flores & Negreiros-Fransozo, 1999*), and varying growth rates (*Negreiros-Fransozo, Costa & Colpo, 2003*), are possible reasons for such a distribution. It has been observed that such a phenomenon is common for organisms that reproduce multiple times in a year and give multiple clutches each season (*Zimmerman & Felder, 1991*). With continuous recruitment and stable larval death rates, the breeding season of unimodal populations can often occur at any time of the year. This is a regular occurrence in populations of tropical fiddler crabs (*MacIntosh, 1989*; *Litulo, 2005a*; *Litulo, 2005b*). On the other hand, bimodality shows seasonality in reproductive events (*Yamaguchi, 2001*; *Costa & Negreiros-Fransozo,*

*2002*). Changes in biotic and abiotic factors like wind pattern, wave action, and nutrient concentration (*Hails & Yaziz, 1982*), as well as changes in the slope of the beach and lunar tidal height (*Clayton & Al-Kindi, 1998*), could affect the size and frequency distribution of such seasonal differences.

The ambient temperature ranged from 27.2 °C to 34.3 °C, with the average temperature being 30.17 °C, suggesting that the study area falls in a tropical climate that can support continuous reproduction (*Patel, Vachhrajani & Trivedi, 2023*). Hence, the occurrence of ovigerous females was recorded in all the months of the year, suggesting that *D. blanfordi* breeds continuously, where the maximum frequency was recorded from August to December. Some other dotillid species have also shown such patterns including *D. fenestrata* (*Litulo, Mahanjane & Mantelatto, 2005*), *D. myctiroides* (*Hails & Yaziz, 1982*), and *D. sulcata* (*Clayton & Al-Kindi, 1998*), as well as *Deiratonotus japonicus* (*Oh & Lee, 2020*), and *Scylla olivacea* (*Ali et al., 2020*) having continuous breeding pattern with peaks in the occurrence of ovigerous females in certain months. Studies suggest that in tropical regions, reproduction occurs almost throughout the year, while in some cases, seasonality is also observed, which is affected by the temperature and rainfall (*Flores & Paula, 2002*). A continuous breeding pattern indicates the production of multiple broods in a breeding season, or the species is breeding asynchronously (*Litulo, Mahanjane & Mantelatto, 2005*), which can also be applied to *D. blanfordi*. On the other hand, the seasonal occurrence of ovigerous females and larval stages might be the result of food availability (*Sastry, Vernberg & Vernberg, 1983*) or strategies to avoid intraspecific competition for food (*Morgan & Christy, 1995*).

In a study carried out on *D. blanfordi* in Karachi, Pakistan (*Manzoor et al., 2016*), the ovigerous females were observed occurring only for six months (February to July), showing seasonality in breeding. However, such variation in the samples could be due to sampling artefacts, as the samples were collected by digging from the substratum only up to 10 cm deep using hand and forceps. According to a standard protocol by *Saher & Qureshi (2010)*, excavations up to 30 cm should be carried out and sieved (1 mm mesh). Hence, adopting inadequate methodology for sample collection might be the possible reason leading to differing results in the population structure of *D. blanfordi* in Karachi, Pakistan, with tropical climatic conditions, which is not far from the present study site. Furthermore, no trend was observed among the ambient temperature and the frequency occurrence of the ovigerous females.

However, it has been found that the juvenile recruiting season was immediately followed by the reproductive season, with a peak occurring during the reduced reproductive time as the prevalence of juvenile individuals dropped with an increase in the occurrence of ovigerous females. According to the present findings, *D. blanfordi* reproduces rapidly and possesses a shorter incubation period, which allows for the year-round recruitment of juveniles. Various other studies have also reported similar phenomena in other species, including *D. japonicus* (*Oh & Lee, 2020*), *D. fenestrata* (*Litulo, Mahanjane & Mantelatto, 2005*), *D. sulcata* (*Clayton & Al-Kindi, 1998*), *Scylla olivacea* (*Rouf et al., 2021*), and *Clibanarius rhabdodactylus* (*Patel, Vachhrajani & Trivedi, 2023*). Several factors, like food availability for adults (*Goodbody, 1965*), larval ecology (*Reese, 1968*), mating, gonadal

growth, incubation period, and time required to attain sexual maturity (*Sastry, Vernberg & Vernberg, 1983*), as well as sea water temperature (*Lancaster, 1990*), could affect the periodicity of the reproductive season. Differences in the reproductive peaks across populations may vary due to various abiotic and biotic factors such as water temperature (*Chou, Head & Backwell, 2019*), salinity (*Huang et al., 2022*), nutritional status of the females, changes in photoperiod (*Zhang et al., 2023*), abundance and availability of food (*Viña Trillos, Brante & Urzúa, 2023*), as well as predation pressure (*Touchon, Gomez-Mestre & Warkentin, 2006*).

It was observed that the egg mass weight, number of eggs, and size of eggs had a positive relationship with the CW and BW of *D. blanfordi*. Similar results were observed in numerous other studies (*Crowley et al., 2019*; *Hamasaki, Ishii & Dan, 2021*; *Aviz et al., 2022*; *Mustaquim, Khatoon & Rashid, 2022*; *Patel, Vachhrajani & Trivedi, 2023*). It was also observed that the ovigerous females having the same CW had differences in egg mass weight, number of eggs, and egg size maybe due to differences in the availability of food, variation in egg production, and loss of eggs (*Hines, 1982*).

The fecundity of *D. blanfordi* was very high in the present study as compared to the previous estimation in Karachi, Pakistan (*Manzoor et al., 2016*). In the case of brachyuran crabs, their fecundity can vary among intraspecific ovigerous female individuals of a similar area or in different regions of the same area, as the fecundity is governed by various intrinsic as well as extrinsic factors. Intrinsic factors such as variation in overall female size, nutrition availability, age of sexual maturity, etc. could lead to differences in fecundity (*Zairion et al., 2015*). While the extrinsic factors include inter- and intra-specific competition. The trade-off between energy expenditure on somatic growth and egg production might affect fecundity (*Zairion et al., 2015*). Additionally, females with a larger CW produce a greater number of eggs, suggesting that CW is one of the main factors for variability in fecundity (*Muiño, 2002*).

## CONCLUSIONS

The present study was aimed at understanding the population structure and breeding biology of *D. blanfordi*. It was observed that there was significant sexual dimorphism, with females being smaller than males, probably because males use their energy for somatic growth while females have to invest in egg production. It was also observed that the overall and monthly populations were male-biased (1:0.88). Such bias could be due to differences in biology and behaviour, as well as the effects of abiotic and biotic factors on males and females. The occurrence of ovigerous females throughout the year suggests that the population is continuously breeding and shows an inverse relationship with the peak in juvenile recruitment, which is commonly observed in other tropical brachyuran crabs. The egg mass weight, the number of eggs, and the size of the eggs all had a positive correlation with the morphology of ovigerous females. Various intrinsic and extrinsic factors, including energy expenditure on somatic growth and egg production, could be affecting fecundity.

## ACKNOWLEDGEMENTS

The authors are thankful to the Department of Life Sciences, Hemchandracharya North Gujarat University, Patan, Gujarat, India, for providing the lab and other facilities. All the authors are thankful to Dhruva Trivedi for technical assistance in field work.

### Funding

The authors received funding from the Researchers Supporting Project number (RSP2024R165), King Saud University, Riyadh, Saudi Arabia. The funders had no role in study design, data collection and analysis, decision to publish, or preparation of the manuscript.

### Grant Disclosures

The following grant information was disclosed by the authors:
The Researchers Supporting Project number (RSP2024R165), King Saud University, Riyadh, Saudi Arabia.

### Competing Interests

The authors declare there are no competing interests.

### Author Contributions

- Krupal Patel conceived and designed the experiments, performed the experiments, analyzed the data, prepared figures and/or tables, authored or reviewed drafts of the article, and approved the final draft.
- Heris Patel conceived and designed the experiments, performed the experiments, analyzed the data, prepared figures and/or tables, authored or reviewed drafts of the article, and approved the final draft.
- Daoud Ali performed the experiments, analyzed the data, authored or reviewed drafts of the article, and approved the final draft.
- Swapnil Gosavi conceived and designed the experiments, performed the experiments, analyzed the data, prepared figures and/or tables, and approved the final draft.
- Nisha Choudhary analyzed the data, authored or reviewed drafts of the article, and approved the final draft.
- Virendra Kumar Yadav conceived and designed the experiments, prepared figures and/or tables, and approved the final draft.
- Kauresh Vachhrajani conceived and designed the experiments, performed the experiments, analyzed the data, prepared figures and/or tables, and approved the final draft.
- Ashish Patel analyzed the data, authored or reviewed drafts of the article, and approved the final draft.
- Dipak Kumar Sahoo conceived and designed the experiments, prepared figures and/or tables, authored or reviewed drafts of the article, and approved the final draft.

- Jigneshkumar Trivedi conceived and designed the experiments, performed the experiments, analyzed the data, prepared figures and/or tables, authored or reviewed drafts of the article, and approved the final draft.

## Data Availability

The morphological parameters of different genders and life stages are available in the Supplementary File.

## Supplemental Information

Supplemental information for this article can be found online at http://dx.doi.org/10.7717/peerj.17065#supplemental-information.

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
