# Peer review of "On population structure and breeding biology of burrowing crab Dotilla blanfordi Alcock, 1900"

_PeerJ, doi:10.7717/peerj.17065_

## Round 0.1 · original submission · Major Revisions

Thank you for your submission to PeerJ. All three reviewers pointed out issues with the manuscript as written that detract from a reader's ability to clearly understand the work carried out and thus the ability to fully critically analyze your work. I agree with such assessment. I recommend having someone proficient in English and familiar with the subject matter review your manuscript. Additionally, revision of the introduction and discussion to improve focus and clarity are necessary. When doing so, I recommend stating clear hypotheses and objectives as two reviewers identified this as an area where the manuscript was lacking as written. Other important issues to address include missing references to figures, limiting references cited in instances (as identified by a reviewer there is one line with 20 citations), as well as figure quality and consistency in the axis scales.

The novelty of the study or potential impact is not at issue, given PeerJ's focus on "methodological soundness" over "impact or novelty;" however, improving the readability of the manuscript is critical to address the methodological soundness of the work presented.

**Language Note:** The Academic Editor has identified that the English language must be improved. PeerJ can provide language editing services - please contact us at [email protected] for pricing (be sure to provide your manuscript number and title). Alternatively, you should make your own arrangements to improve the language quality and provide details in your response letter. – PeerJ Staff

Reviewer 1 ·

Basic reporting

The English language should be improved to ensure that an international audience can clearly understand your text. For example - the language used in lines 52, 100, 118, 119, 120, 209, 228 - the current phrasing makes comprehension difficult. I would suggest having someone proficient in English and familiar with the subject matter review your manuscript. I found your literature references and background sufficient. The resolution of Figure 1 does not show the funnel shaped coast you refer to in the text. I couldn't find a reference to Figure 2 in the text so it should be omitted. In Figure 3, it indicates 60 meters between each quadrat when it should say 50m. The raw data is useful and easily understood.

Experimental design

I question the sampling of the water from pools nearby as opposed to sampling the seawater as stated in line 139. A tidal pool would fluctuate with ambient air temperature and therefore simply be another measure of the ambient air, whereas the seawater would have been a better piece of information. Therefore, I would remove this from the materials and methods, results and discussion.

The statistical analyses of size was confusing. I would suggest a comparison between males and all females as this would validate the dimorphism between males and females. Then compare the ovigerous to non-ovigerous females separately.

Validity of the findings

This study is interesting and expands the knowledge of a semi-terrestrial brachyuran species from a unique local.

Reviewer 2 ·

Basic reporting

The grammar is ok, but there are some few cases that need improvement .
The literature references are educated
The introduction need improvement, see comments in the section below
There are two figures that could be mixed in one, see comments in the section below

Experimental design

The research question is not well defined, and do not fill any knowledge gap. However their primary objective was measure population structure of a crab populations, and was well executed. The methods are described well, there are some minor details that can be easily fixed .

Validity of the findings

There is no potential impolitic of these findings, and just provide basic information . With no real scientific question behind. Do not feel any gap of scientific knowledge . Methods and statistics are ok and educated to their objectives. The discussion is not well written, there is redundant and no connected information that probably goes beyond to support their results.

Additional comments

In the paper titled "On population structure and breeding biology of the burrowing crab Dotilla blanfordi Alcock, 1900," the authors conducted an extensive study aiming to measure the population structure and breeding biology of the burrowing crab D. blanfordi in the sand beach of Bhavnagar, located on the Gulf of Kachchh, Gujarat coast, India. The authors' work spanned approximately one year. The methods employed were appropriate and aligned with their objectives, resulting in reliable findings. However, there are several issues with the writing. For instance, the abstract lacks a clear indication of the importance of the study and the underlying hypotheses. The introduction provides basic information on population structure and includes some details about the Dotillidae family. However, the information presented is ambiguous, repetitive, and overly descriptive. It lacks a specific research question of interest and a clear hypothesis. The discussion section is excessively lengthy, containing unnecessary and repetitive information that could be omitted to make it more concise.
Research on population structure and breeding biology is common, and this particular study does not contribute any novel insights to the field. There is no clearly defined hypothesis or unique problem being addressed.


Specific comments

Abstract

Lines 33 and 34 you are saying basically the same. …Is commonly found in the intertidal region of Gulf of Kachchh, Gujarat and was studied on the sandy beach of Bhavnagar located on the Gulf of Kachchh…

Line 61-62: This sentence does not connect with the main idea and does not flow naturally.

Lines 63-64: The idea presented in this sentence is not clearly written, and it is difficult to understand the intended meaning. However, it could be rephrased as follows: "Furthermore, estimating population fecundity and sexual maturity is crucial in predicting the turnover capacity of natural populations and understanding the long-term impact of the environment."

Lines 73-75: Including a large number of references is not necessarily a problem, but 20 references in three lines may be excessive. It is important to check the journal's regulations and follow the guidelines for the number of references allowed.

Lines 81-83: The information presented in these lines does not seem to be directly related to population structure. It may be better to either remove this information or find a way to connect it more explicitly to the main topic of the paper.


Lines 91-92. This basically the same you say in the lines 75-77

Lines 92-97 this is similar to what you say in the beginning of the introduction.

Lines 92. species’ is not scientific writing

LINES 103-105. Most of the population structure of crab species around the world has not been investigated. However I think think that this is sufficient justification for the research presented in the paper. It is essential to provide a more specific research question or problem of interest to justify the study. Is there any relevant question of internes behind this research?


M & M

Line 112. I think is better avoid the use of “viz”. Many international readers will not understand this expression and is confusing

Line 113. Monsoon is is type of weather phenomenon characterized by seasonal wind patterns and changes in precipitation, is not a season.


Line 114. This value of temperature is confusing, is what you got in your measurements? If yes, should to go results. If not, it needs a reference.

Line 117. …the highest tidal range in the Arabian Sea and second largest measured anywhere in the world. You need a reference for this.


Line 119. Very less" is not grammatically correct. Instead, you can use "very minimal" or "very low" to convey the intended meaning.

Line 119-125. This should go to results

Lines 109- 125. Actually, this whole paragraph is so confusing. There is a lot of information and data. There are not references and some parts apparently should be in results, or you got them from another paper?

Lines 132-142: The information presented in these lines is mixed, making it difficult to follow the flow of the description. The paragraph begins with a description of the transects, then jumps to information about the crab, and then to the quadrants. This mixing of information can be confusing for the reader.

Line 137. This line is not well written: the quadrate area was excavated up to 30 cm since the species do not burrow beyond that and sieved. Something like this? The quadrate area was excavated up to 30 cm since the species does not burrow beyond that depth, and the soil was sieved.

Line 151. 40 ovigerous female each month?

Line 160. Change “which was 10” for (DL=10)

Line 173. Gender or sex?

Line 186. change investigated for estimated.

Results

Lines 192-194. maybe add averages and error for those values

Line 207. Add comma after CW. …at 3.4 mm CW, and 1.2 mm…

Line 218. In which figure we can see that, Fig 6?

Line 222. What is the P value for that significance?

Discussion.

Line 224-225. The whole sentence is too large

Line 225. I don’t see the need to bring the Searmidae family here

Lines 224-232. Actually, the whole sentence is not related with your research

Lines 233-241. You start talking about comparation of sizes between males and females, then the line 237 you jump to some random explication that actually, is not explain why you got bigger females. Then you talk about chances of males attracting females. I think all that in the lines 237 to 241 should be deleted. Its reads as random information no connected with the fact that you got bigger females. Actually, that should be connected with the information in lines 242.

Lines 243 to 247. As you have many examples of carb species, is clear that is a pattern of having more males during the year. Is there any research where you find the opposite results? That would be interesting for the discussion.

Also, in several parts of the discussion you have, “this have also been observed in XX, xx, xxx, and many more species”. This is too repetitive and unnecessary. And, this also makes me think that results that have been proven in so many species, that is actually not of the interest of a high impact journal.

Line 363. According to the direction of the journal, funding is not for the acknowledgment section.

Figures

Fig 3. Looks like the maps is backwards. I mean, the mark of 0 meters shouldn’t be next to the sea? I don’t think this figure is really need it. Maybe you should delete this figure and mixt it with figure 1. Delete Fig 1A and put the Fig 3 instead.


Figure 4. y axis should be both in the same scale for both species, between 0-120 .

·

Basic reporting

1. The Results section and Figures need significant work.
a. Figure 3 is never referenced in text.
b. If a statistical test was run, particularly for significant interactions, need to be written in text or presented in a table that is referenced. The authors do this for some analyses but specifically nothing is presented for the linear models presented in Figure 7.
c. Figures 4 and 5 would look nicer presented as stacked bar plots since it is showing a proportion of individuals. Further, please limit the y-axis to 0 – 100 as you cannot have 120% of individuals.
d. Please remove trend lines in Figure 7 for non-significant interactions.
e. Line 213: There is not a statistically significant interaction here, please do not interpret the results as a “positive correlation”. This should be avoided throughout.
f. I do not understand the utility of Figure 6 – further, the use of dual y-axes is not considered best practice for data visualization. Please reconsider the figure as is or provide a clearer justification for its inclusion.
2. The authors in general assume the readers have a background in crustacean biology. For example, there is no explanation of the terms brachyuran, anomuran, or ovigerous. The journal is open to a wider audience who may or may not be familiar with such terminology. A few, brief words to define such terms would greatly increase readability.
3. The authors should get the manuscript checked for English language as there are a multiple word choice/use and grammar issues that takeaway from the reading. Examples, Line 52: “biology majorly try” and Line 100: “species to species of population to population”.
4. Lines 101-107: Please explicitly state the objectives and hypotheses for the study.
5. Figures 1 and 2 are never referenced in text.
a. Lines 130-131 would be a good place to reference Figure 2.
6. There are several sections of extraneous citations, ex: Lines 235-237, 243-247, and 274-277. Providing support for your findings in the literature is good practice, however too much can hinder readability. Perhaps limit to the most recent or relevant?

Experimental design

No comment.

Validity of the findings

No comment.

Additional comments

7. Line 292: the temperature range does not seem to fit with the average temperature of 21.9C in Line 114.
8. Line 353: There is an assumption made several times in the text that males invest more in growth than females with no citation or supporting evidence. Please correct this or temper the statement.

---

## Round 0.2 · Minor Revisions

Dear authors,

All three reviewers appreciate the changes made to improve the quality of the manuscript. Two reviewers provide several suggestions that although minor I feel need to be incorporated prior to acceptance. Please read through the feedback provided by reviewers #1 and #2 and incorporate or address these suggestions in a rebuttal letter. For reviewer #2, please take a look at the feedback they provide in the annotated manuscript they provided.

Cheers,
Carlos A. Santamaria, Ph.D.

Reviewer 1 ·

Basic reporting

No comment

Experimental design

The delineation of size classes determined to be juvenile seems arbitrary, I would recommend identifying previous studies that utilize a similar method to classify juvenile. In this study, it appears your results are a direct artifact of this classification scheme.

Validity of the findings

The only area that is questionable is the relationship between ovigerous females and juvenile abundance. Because the size of juveniles was determined by the smallest ovigerous female, there will always be a direct negative relationship with any shift in one class resulting in the opposite shift in the other.

Additional comments

Edits:
Line 30 – remove “crab” from between burrowing and brachyuran.
Line 69 – should it say “pellets” instead of pallets?
Line 104 – remove the rest of the sentence after D. blanfordi.
Line 186 – change showing to showed.
Line 188 – change to “dominated by the species Ocypode rotundata.
Line 189 – italicize D. blanfordi
Line 194 – change to “We observed prominent sexual dimorphism”
Line 201 – remove “the” at the end of the line.
Line 202 – add “the” after during
Line 215-221 – Regarding the discussion on relationship between frequencies of ovigerous females and juveniles. Of course you see a direct relationship between the two because you defined juveniles as those individuals smaller than the smallest ovigerous females. If the cohort you identified as juveniles in the first three months of your study molted, then they would be larger and thus of the size you considered non-juveniles. I suggest removing the last line of the paragraph and stopping at (Fig.6) on line 218.
Line 225 – was the relationship positive or negative?
Lines 228-232 – Not sure the relevance of this paragraph to the discussion. It is more fitting in the introduction to emphasize the need to study this species.
Lines 298-299 – Clarify the end of this sentence, as it reads as an incomplete sentence. Did the other species have similar breeding peaks during the same months?
Line 300 – remove “majorly”
Line 304- change Ovigerous to ovigerous
Line 316 – among the ambient what?
Line 317 – See the note above regarding lines 215-221.
Line 337 – edit the sentence. Do you have evidence of a difference in the availability of food, etc? If not, then add “may be” in front of due.

Reviewer 2 ·

Basic reporting

no comment

Experimental design

no comment

Validity of the findings

no comment

Additional comments

Dear authors

Im really happy with new version. I have some minor changes that I consider you should do mostly in the introduction . See the attached document

Cheers!

Annotated reviews are not available for download in order to protect the identity of reviewers who chose to remain anonymous.

·

Basic reporting

Any issues with basic reporting from the first round have been addressed satisfactorily.

Well done.

Experimental design

The experimental design is sound. Any issues have been cleared up since the first round of review.

Validity of the findings

Findings are valid and interesting. They provide a nice addition to the literature surrounding small crabs.

---

## Round 0.3 · accepted · Accept

I want to thank you for taking the time to address all the comments, edits, and suggestions the three reviewers have made throughout the review process. I believe the manuscript is ready to be accepted for publication. I request the authors make the following changes during the production process:

Line 85: italicize "blanfordi"
Line 153: should say "analyses"
Line 217: ",- " appears to be a typo
Line 247: "relation" should read "correlation"
Line 311: continual or continuous?
Line 329: "isaffected" should say "is affected"
Line 401: "Fundings" should say "Funding"